# Dynamic allometry in coastal overwash morphology

Eli D. Lazarus[1*], Kirstin L. Davenport[1], Ana Matias[2]

[1]Environmental Dynamics Lab, School of Geography and Environmental Science, University of Southampton, Highfield B44, Southampton, SO17 1BJ, UK

[2]Centre for Marine and Environmental Research, University of Algarve Campus of Gambelas, Faro, Portugal

Correspondence to: Eli D. Lazarus (E.D.Lazarus@soton.ac.uk)

**Abstract.** Allometry refers to a physical principle in which geometric (and/or metabolic) characteristics of an object or organism are correlated to its size. Allometric scaling relationships typically manifest as power laws. In geomorphic contexts, scaling relationships are a quantitative signature of organisation, structure, or regularity in a landscape, even if the mechanistic processes responsible for creating such a pattern are unclear. Despite the ubiquity and variety of scaling relationships in physical landscapes, the emergence and development of these relationships tend to be difficult to observe – either because the spatial and/or temporal scales over which they evolve are so great, or because the conditions that drive them are so dangerous (e.g., an extreme hazard event). Here, we use a physical experiment to examine dynamic allometry in overwash morphology along a model coastal barrier. We document the emergence of a canonical scaling law for length versus area in overwash deposits (washover). Comparing the experimental features, formed during a single forcing event, to five decades of change in real washover morphology from the Ria Formosa barrier system, in southern Portugal, we find differences between patterns of morphometric change at the event scale versus longer time scales. Our results may help inform and test process-based coastal morphodynamic models, which typically use statistical distributions and scaling laws to underpin empirical or semi-empirical parameters at fundamental levels of model architecture. More broadly, this work dovetails with theory for landscape evolution more commonly associated with fluvial and alluvial terrain, offering new evidence from a coastal setting that a landscape may reflect characteristics associated with an equilibrium or steady-state condition even when features within that landscape do not.

## 1 Introduction

In geomorphology, a scaling law is a formalised expression that typically describes how two geometric attributes of a landform relate to each other in a consistent way. Most geomorphic scaling laws take the form of a power relationship. For example, the length ($L$) of a feature relative to its area ($A$), as in a fluvial drainage basin (Hack, 1957; Montgomery and Dietrich, 1988), is typically expressed $L \sim A^h$, where the scaling exponent $h$ defines the slope of the relationship in log-transform space. Geomorphic scaling laws derived from feature dimensions demonstrate allometry: a general physical principle in which geometric (and/or metabolic) characteristics of an object or organism are correlated to its size. Allometric

patterns appear in a diversity of geomorphic settings – erosional (river systems, submarine canyons) and depositional (alluvial fans, coastal deltas) – and are a quantitative signature of intrinsic structure, organisation, or regularity (Church and Mark, 1980; Dodds and Rothman, 2000; Moscardelli and Wood, 2006; Straub et al., 2007; Paola et al., 2009; Wolinsky et al., 2010; Edmonds et al., 2011; Lazarus, 2016). Scaling relationships that describe geomorphic allometry can serve as useful predictive tools, even when the processes behind the patterns are complex or unclear (Shreve, 1966; Kirchner, 1993).

Whether, and how, power laws might provide explanatory insight into physical mechanisms has been a recurrent issue in different academic fields for the better part of the last century. A flurry of critical discourse on the utility of geomorphic scaling laws unfolded in disciplinary literature of the 1970s. Following Horton's (1945) pivotal observations of mathematical structure in stream networks, a variety of geomorphic scaling relationships gained traction both in drainage (Langbein et al., 1947; Leopold and Maddock, 1953; Hack, 1957; Strahler, 1957, 1958; Melton, 1958; Gray, 1961; Leopold et al., 1964) and alluvial settings (Bull, 1962; Denny, 1965). Borrowing primarily from biology (Huxley, 1924), but also from urban geography (Berry and Garrison, 1958) and economics (Simon and Bonnini, 1958), Woldenberg (1966) applied the concept of allometric growth – defined as "growth of a part at a different rate from that of a body as a whole" (Huxley and Tessier, 1936) – to the emergence of stream order within a drainage basin (Horton, 1945). This extension of allometric theory triggered an essential, not simply semantic, terminological distinction between *allometry* versus *allometric growth* (Mosley and Parker, 1972). The broadest conception of allometry includes all relationships that describe "a size-correlated change in shape" (Gould, 1966). Allometric growth, meanwhile, implies that the *rates* of size-correlated changes in shape describe an organised relationship. A test of drainage-network evolution in an experimental basin by Mosley and Parker (1972) did not confirm Woldenberg's (1966) supposition of allometric growth: they found no evidence that rates of size-correlated changes in drainage networks conformed to a well-organised pattern. However, the description of their null result did not disparage geomorphic allometry as a subject for further research. Rather, Mosley and Parker (1972) describe their test of Woldenberg's (1966) theorising with a tone of generosity that further opened the problem, encouraging geomorphologists to consider the inductive challenges posed by "static" versus "dynamic" allometry (Bull, 1975, 1977; Church and Mark, 1980).

The majority of geomorphic scaling laws exemplify static allometry: snapshots of landform examples sampled from a collection of different sites, or from a "population" of many examples at a single site. Aggregating examples in this way enables objective comparison across a variety of cases and contexts – for example, compiling field data from different physical environments or from new and historical observational records, or placing field observations and model results in the same parameter space. But where static allometry reflects the "interrelations of measurements made of an object at one time in its history", dynamic allometry reflects sequential measurements of shape over time (Bull, 1975). Linking the static allometry of a given shape to its dynamic allometry – observing the progression by which a form emerges – is fundamental to linking overarching pattern to underlying process.

Depending on the landform of interest, directly observing dynamic allometry in real settings can be extraordinarily difficult:
consider, for example, that any feature dominated by diffusion will change on a time scale proportional to the square of the feature's length scale, $t \sim L^2$. (That is, the bigger the landscape feature, the more patient an observer needs to be.) Other features, like event-triggered fluvial (Millard et al., 2017), coastal (Leatherman and Zaremba, 1987; Lazarus, 2016), and other mass-transport deposits (Moscardelli and Wood, 2006) may be too violent or dangerous to track dynamic allometry *in situ*. Deposits in the geologic sedimentary record are fossils (Moscardelli and Wood, 2006; Donnelly and Woodruff, 2007), long since active – although glimpses of their past dynamics might be possible to infer from fine-scale stratigraphy, if well enough preserved (Shaw et al., 2015). From the comparative safety of controlled experimental or numerical model systems, time scales of physical landscape change are deliberately accelerated, and dynamic allometry can be revealed in high-frequency time series.

Here, presenting new evidence of dynamic allometry from a previously reported physical experiment on coastal barrier overwash morphology (Lazarus, 2016), we document the progressive development of (1) a stable scaling exponent relating length and area in washover deposits, and (2) a dominant aspect ratio (or "spacing ratio") relating washover width to length (Fig. 1). Overwash is a natural, fundamental physical process of coastal barrier systems in which shallow overland flow, driven by a storm event, transports sediment from the open-coastal barrier face to the barrier floodplain and sheltered back-barrier wetlands (Morton and Sallenger, 2003; Donnelly et al., 2006; FitzGerald et al., 2008). Overwash occurs even in the absence of sea-level rise, but sediment supply to the floodplain and back-barrier environments through washover deposition is the vital mechanism that allows barriers to maintain elevation and width relative to sea level over time scales of centuries to millennia (FitzGerald et al., 2008). Although essential to coastal barrier evolution (and, by extension, to the function of natural barrier ecosystems), overwash becomes a hazard where it interacts with coastal infrastructure and built environments (Rogers et al., 2015; Lazarus and Goldstein, 2019). Overwash morphology is thus at the crux of understanding current – and anticipating future – coastal environments and risk along low-lying open coastlines (Wong et al., 2014).

From an ensemble of experimental trials that each simulated a single storm event, we measured dynamic allometry in the "population" of washover features that formed across a generic, spatially extended experimental domain. We also investigated allometry in a sample of individual washover features that we tracked through time as they developed. Furthermore, to begin exploring signatures of dynamic allometry over significantly longer time scales in a real setting, we examined repeated measurements of overwash morphology along the Ria Formosa barrier system, in southern Portugal, from aerial images spanning five decades (Matias et al., 2008). We find a motivating correspondence between the experimental and real cases, and our results complement explorations of dynamic allometry in cognate geomorphic systems (Perron and Fagherazzi, 2012). Given that leading process-based models of coastal morphodynamics have embedded in their architectures a host of semi-empirical parameters (Simmons et al., 2019), scaling relationships derived from static and

dynamic allometry for overwash morphology may first serve tests of model predictions, en route to integration into predictive models themselves.

## 2 Data and methods

### 2.1 Physical Experiment

Experimental data comes from orthorectified overhead images of a physical model of a coastal barrier that produced spatial sets of washover deposits formed by overwash flow. The original experiment is detailed in Lazarus (2016). The essential design consideration was geometric: a low barrier (a small difference in relative height between the barrier top and back-barrier platform) with an extended aspect ratio in the alongshore dimension. The experiment was conducted in a sediment tank (500 x 300 x 60 cm) at St Anthony Falls Laboratory (National Center for Earth-surface Dynamics, Minnesota, USA). Using well-sorted, coarse river sand (D50 ~0.59 mm), a low, rectangular, topographically smooth barrier with an extended aspect ratio (90 x 300 x 2 cm) was constructed across the tank, creating an "ocean" reservoir on one side and a level back-barrier plane on the other.

To run a trial, the water level in the reservoir was raised with a constant infill rate. When the reservoir water level exceeded the barrier height, flow travelled in a continuous front across the barrier top and down onto the back-barrier plane, incising overwash throats and depositing washover lobes along the back-barrier edge. The only hydrodynamic forcing came from water height relative to the barrier, making the experiment broadly representative of an "inundation regime" in the hierarchy of extreme storm impacts (Sallenger, 2000). Over tens of minutes, competition for available flow meant that overwash morphology developed at different rates along the barrier. Each trial ran until the barrier reached effective steady state, when little or no sediment movement was evident.

Flow drained at the far boundary of the back-barrier plain. Overwash flow over the barrier was observed to be subcritical, at a flow depth $\sim\leq1$ cm, but was locally supercritical within the throats; in the field and in large-scale experiments, cross-shore overwash flow tends to be supercritical (Matias et al., 2010, 2016). Despite subtle indications of percolation through the back-barrier face, throats only formed in response to overwash flow, not from back-barrier slope failure or groundwater sapping. Each of the trials reported here used the same barrier height (2 cm) and infill rate (~0.3 Ls$^{-1}$). Washover size could be increased by increasing the barrier height. A height of 2 cm was chosen because that elevation generated approximately five times more overwash features alongshore than a barrier with height 4 cm (see Supporting Information for Lazarus, 2016). The infill rate was the maximum possible for the experimental basin, and not so powerful that it would simply obliterate the initial barrier (since a catastrophic storm was not the intention of this experimental design). Although grain size was not directly tested as a control on experimental overwash morphology (working to time and labour constraints, we used the sand that was already installed in the basin at the time of its availability), we inferred that a larger grain size would likely

result in blunter lobes, and a finer grain size in more "finger-like" deposits (Homsey, 1987) significantly greater in cross-shore length relative to their alongshore width.

From overhead camera imagery, we used digital geospatial software to measure washover morphometry: length $L$, in the cross-shore dimension; width $W$, in the alongshore dimension; area $A$; and alongshore spacing between washovers. (Previous work by Lazarus (2016) used laser-scanned topography of the final experimental condition.) Experimental washover features were approximately 10–20 cm long ($L$) in the cross-shore dimension (Fig. 1).

### 2.2 Field examples spanning multiple decades

Measurements of washover morphology from the Ria Formosa barrier system in southern Portugal are described in detail by Matias et al. (2008). The system includes a group of seven sandy barriers that wrap around a pronounced right-angle bend in the coastline (from approximately northwest–southeast to southwest–northeast), capturing two different wave exposures (to the southwest and southeast, respectively). Washover length and area were recorded from sets of orthogonal aerial photographs taken between 1952 and 2001 for a total of 369 washover sites. A subset of persistent washover sites were found in up to four images spanning multiple decades. Washover length (cross-shore distance between barrier crest and back-barrier edge) in the Ria Formosa data reached a maximum of 250 m. Barrier morphology varies substantively within the Ria Formosa system, ranging by an order of magnitude in island width (in some cases along the same island), with differing patterns and extents of dune fields, urban footprint, and proximities to mesotidal inlets. Here we examine the Ria Formosa not because of any direct correspondence to the barrier design in the physical experiments, but because the system offers a closely examined source of repeated measurements of persistent washover footprints along its ~60 km spatial extent. Note that Figure 1, adapted from Lazarus (2016), shows washover measurements from Core Banks, North Carolina (USA), and a globally distributed dataset from Hudock (2013) and Hudock et al., (2014). Both datasets represent "snapshots" of population allometry and help motivate this work, but we do not re-analyse them here.

### 3 Results

Previous comparative analysis of field and experimental observations of overwash morphology demonstrated similarity in morphometric scaling (Lazarus, 2016), but looked only at "final" landform configurations (Fig. 1). Here, we present new analyses that examine the evolution of overwash morphology. We aggregated measurements from an ensemble of experimental trials and series of aerial images, respectively.

### 3.1 "Population" allometry

We confirm at the outset that no single trial nor imagery year determines the ensemble scaling relationships that we calculate (Fig. 2a and 2b), nor do other contextual variables, such as barrier segment or orientation, appear to affect the collective

scaling in the field measurements (Fig. S1). The experimental ensemble scaling relationship between $L$ and $A$ shown in Figure 2a and 2b, generated from overhead imagery, yields a smaller scaling exponent $h$ than the equivalent relationship reported by Lazarus (2016), generated from three-dimensional topographic laser scans (Table 1). However, the 95% confidence intervals around the respective scaling exponents show reasonable agreement between the different experimental measurements (from overhead imagery versus laser topography), and imagery-based measurements from the experiment and Ria Formosa show closely corresponding scaling exponents (Fig. 2; Table 1).

A sequence of images from an experimental trial illustrates how the scaling exponent relating washover length to area, for a "population" of related washovers, changes over time during a single forcing event, also capturing the emergence of a dominant width-to-length aspect (or spacing) ratio between washovers alongshore (Fig. 3). Aspect ratio – defined as alongshore width relative to cross-shore length, or $W$:$L$ – can be calculated for any individual washover. When washovers are arrayed alongshore, such that neighbouring features share an edge, the aspect ratio reflects a normalised measure of spacing between adjacent deposits (Fig. 1). The same principle holds for contiguous drainage basins arrayed along quasi-linear mountain fronts (Hovius, 1996; Talling et al., 1997; Perron et al., 2009). Here, experimental washovers were initially long and narrow, yielding a high length-to-area scaling exponent (a steeply sloping fit in log-log space) and a low width-to-length aspect ratio. As a trial progressed, washovers rapidly widened – and in many cases, merged – growing in area relative to length, causing the length-to-area scaling exponent to decrease and the aspect ratio to increase.

Measuring the length-to-area scaling exponent and aspect ratio from sequential images in each experimental trial yields the evolution of these scaling metrics as a function of per cent trial run-time (Fig. 4). The populations in each trial converge on dominant scaling relationships. Because these scaling relationships are quantitative signatures of a predominant morphological expression and preferred spatial arrangement, documenting their spatio-temporal development is a step toward connecting pattern to process. Hypothetically, the "final" scaling relationships could have been the consequence of each washover conforming to singular dimensional constraints from the outset, resulting in a constant scaling relationship through time. Instead, we find convergence toward a dominant pattern signature, indicative of spatial self-organisation (Lazarus and Armstrong, 2015). If washovers are initially too far apart, they widen, and some new deposits fill in between them, until the alongshore pattern reaches a closer spacing configuration. Conversely, if washovers are initially too close, they merge, effectively adjusting their centroids to be farther apart. The spacing ratio converges on a quasi-equilibrium configuration (Fig. 4b) more quickly than the length-to-area relationship does, perhaps because the mean spacing ratio is a more stable metric than the scaling exponent, which is comparatively more sensitive to the influence of larger washover deposits as they grow. That is, a large washover deposit is less likely to markedly shift the centre of a univariate distribution of spacing ratios than it is to affect a best-fit power relationship between length and area.

Also evident in the experimental results are irregular, persistent gaps between washover sites (Fig. 4c), which are likely a consequence of overwash flow competition and partitioning. The infill rate was never varied, meaning the hydrodynamic forcing was held constant throughout each trial. Moreover, the surface of the upstream, "ocean-side" reservoir was never perturbed (agitated with a wave paddle, for example). This means that once enough overwash breaches had formed to accommodate and distribute the forcing flow across the barrier, new breaches were either unlikely to develop, or would only

develop as a consequence of subtle, local interactions between adjacent overwash throats. For example, if flow through a throat slowed down, sediment caught in the throat could form a temporary plug. That plug appeared to drive a backwater effect that elevated the upstream water surface just enough to force the overwash flow toward a new path of steepest descent – typically down through a neighboring throat, but sometimes over an otherwise undissected reach of the barrier. (These plug-and-backwater dynamics were observed, but not measured directly.) Over many tens of minutes, the overwash

morphology adjusted to flow conditions and the elevation difference between the barrier top and back-barrier plane, sets of neighbouring throats might plug and unplug several times, with corresponding periods of dormancy or growth in their associated washover deposits. Given that they were subject to the same forcing conditions, all breaches in the barrier should have tended to adjust toward the same open-channel geometry. However, because throats had to share – compete for – available overwash flow, closely spaced sets of throats grew more slowly than an isolated throat with no nearby neighbors.

In a natural case, flow-limited conditions may mean that for a series of overwash throats, no single throat may ever capture enough flow to reach its equilibrium open-channel configuration. Storm-driven water levels in the field (Shaw et al., 2015; Englestad et al., 2018; Wesselman et al., 2019) may rise and fall much faster than the time scales required for overwash morphology to reach a geometric equilibrium.

**3.2 Dynamic allometry of individual washovers**

The experiments reflect morphometric development in individual washovers during a single forcing event (Fig. 5a). The dynamic allometry of individual washovers shows that their growth trajectories are variable. In general, the washovers all grow over time, moving up the trend of the length-to-area scaling relationship if not necessarily conforming to its slope. Such variability evident at the individual level – likely an indirect reflection of morphodynamic changes occurring

simultaneously to neighbouring washovers alongshore (Fig. 4) – suggests that collective convergence to a predominant scaling relationship is an emergent behaviour of the larger barrier system.

In contrast to the single forcing event of the experiments, measurements of individual washovers at Ria Formosa (Fig. 5b) reflect morphometric changes sustained over multiple decades from many forcing events and long periods of quiescence

(Matias et al. 2008). Over time, the washovers shift both up and down along the length-to-area scaling relationship, suggesting that once established, a preferred spatial configuration may exert significant control on subsequent morphological change. Shifting up along the scaling relationship during a decadal interval – indicating washover growth – suggests overwash site reactivation (Hosier and Cleary, 1977). The same washover might intermittently increase its footprint with

successive storm strikes, even if its total volume might decrease through aeolian deflation (Leatherman and Zaremba, 1987). Reactivation events may also be partial or incomplete, relative to the maximum footprint at a given site.

Perhaps a more surprising result is where washovers shift down the scaling relationship, to a smaller but still dimensionally consistent size. Such dimensional adjustment could stem from depth-dependent zonation in barrier vegetation. (Matias et al. (2008) discuss the potential influence of barrier vegetation on Ria Formosa overwash morphology, but do not measure it directly.) Storm deposits can drive spatial heterogeneity in vegetation growth rates: vegetation buried too deeply by a storm deposit will die, but for some dune and marsh plant species, shallow to moderate burial can stimulate growth (Maun and Perumal, 1999; Gilbert and Ripley, 2010; Walters and Kirwan, 2016). Differential plant response to burial can thus determine spatio-temporal patterns in barrier vegetation cover, as the perimeter of a washover deposit may foster an envelope of new plant growth. Those spatio-temporal vegetation patterns in turn facilitate or inhibit pathways of sediment transport across the barrier (Goldstein et al., 2017). If dynamic allometry informs how washover morphology takes shape, then its growth pattern may also work in reverse, informing zonal patterns of depth-dependent vegetation cover that effectively reduce the dimensions of a washover footprint. Furthermore, gradual processes of vegetation recovery and aeolian sand deposition within the washover can progress in irregular ways related to natural topographic heterogeneity in barrier and dune morphology, forcing the washover to inherit morphometric characteristics dynamically unrelated to overwash and inundation processes. Scaling controls likely manifest in conjunction with, not in place of, other allogenic factors that affect overwash morphology over multi-decadal time scales, such as relative sea-level rise, changes in shoreline position and sediment supply, and heterogeneity in shoreface lithology (Morton and Sallenger, 2003; Matias et al., 2008). Preferred geometric relationships in overwash morphology may both set a template for smaller-scale, faster-forming barrier features and be forced to conform to contextual controls exerted by barrier-scale geography (Werner, 2003; Coco and Murray, 2007).

### 3.3 Allometric growth

We also looked for evidence of allometric *growth* in the time series for the five individual washovers tracked in the experiment (Fig. 5a), comparing *changes* in length to the corresponding *changes* in area. Echoing Mosley and Parker (1972), who found no clear indication of allometric growth in the evolution of an experimental river network, we find no clear indication of allometric growth in washover (Fig. S2). Again, perhaps the apparent absence of any scaling in the rates of change is because these washovers did not grow in spatial isolation, and were instead responding to changes occurring at neighbouring washover sites alongshore.

## 4 Discussion and implications

### 4.1 Extension to related geomorphic systems

The convergence to a quasi-equilibrium alongshore spacing that we observe in the experimental overwash morphology (Figs. 3 and 4) is an empirical complement to numerical experiments demonstrating the emergence of regular spacing in ridge-and-valley topography (Figure 7 in Perron and Fagherazzi, 2012). The physical basis for the regular spacing that emerges in the barrier experiment may be closely related to its upland analogue (Lazarus, 2016). In ridge-and-valley topography, a preferred wavelength arises from spatial competition among incipient drainages for drainage area (Perron et al., 2008, 2009). Valleys with greater area have greater flow capture, enabling them to deepen and propagate headward as slightly larger drainages gain a competitive advantage over slightly smaller neighbours. Neighbouring large valleys expand simultaneously but the drainage area of the divide between them diminishes, ultimately inhibiting further valley growth. Subtle differences in topographic gradients from one valley to the next then determine minor, intermittent changes in drainage area.

Perron et al. (2008, 2009) do not mirror this explanation onto depositional patterns; all material exported from their numerical valleys disappears from the model domain. Still, a kind of reversal of the advection–diffusion mechanism that they describe for ridge-and-valley terrain is plausible for its depositional counterpart. Our analysis only addresses the depositional washover patterns, but Lazarus (2016) documented the full domain of the experimental topography, including the spatial array of source drainages (overwash throats) along the barrier. In the experiments, sediment was initially advected to the back-barrier, especially by the first pulse of inundation over the barrier top, tending to create washovers that were long and narrow (Fig. 3). The washovers then grew laterally and gradually, as flow was forced to spread over them, diffusing fresh sediment to their margins.

Voller et al. (2012) offer a theoretical, analytical explanation for such a reversal in the flow of geomorphic "information" across erosional–depositional transitions, where information moves downstream in erosional settings and upstream in depositional ones. Their construct is one-dimensional, but sets up a two-dimensional thought experiment. In a spatially extended erosional domain, if competition for drainage area is the underlying driver of information downstream, then in a spatially extended depositional domain, competition for accommodation space could be the underlying driver of information upstream. As deposits grow, merge, and avulse into new available accommodation space, which gets more limited as deposition increases, they transmit that information upstream in the form of channel profiles and backwater gradients. That information, in turn, is registered at the base of the sourcing valley, triggering either a reduction or increase in erosional export.

In real barrier overwash conditions there is no upstream drainage-valley source – but there is a zone of onshore hydrodynamic forcing that, hypothetically, could feel a backwater effect or similar change in water surface gradients along the barrier front that triggers interactions among neighbouring overwash sites (Lazarus and Armstrong, 2015). The concept and construction of the experimental barrier examined here may have much in common with a study of rill formation by Izumi and Parker (1995), who show analytically why a free water surface over an erodible plane will develop backwater

effects and Reynolds stresses that inhibit runaway incision of "infinitely narrow, infinitely deep" channels (Perron et al., 2008). The formation of a barrier inlet might be a case in which a positive feedback allows a single cross-shore transport pathway to effectively capture all available forcing flow over a large spatial area. However, at a larger spatial scale, dynamical interactions among groups of inlets suggest the influence of spatial competition for the prism of water created by onshore forcing (Roos et al., 2013).

The dynamic allometry of individual washovers that we record (Fig. 5a) bears on an even broader geomorphological question regarding landscape convergence toward an equilibrium state – and, in particular, how such convergence is physically expressed (Bull, 1975; Perron and Fagherazzi, 2012). Describing a numerical modelling example of self-organized valley spacing, Perron and Fagherazzi (2012) remarked that "different landscape features can have very different response times, and that some can be out of equilibrium even while others appear to be close to a steady state". Bull (1962, 1975, 1977) pursued similar observations for alluvial deposition at the outlet of drainage valleys, and we suggest that the same concept applies to our model system of coastal overwash morphology: we see the population of washovers describing allometric relationships (Figs. 3 and 4), and we see individual washovers reflecting that collective allometry to varying extents.

### 4.2 A consequence of process, not a direct measure

Geomorphic scaling laws are typically constructed from well-developed, steady-state topography, or from a broad sample of isolated landforms. Opportunities to record stages of dynamic allometry in a landscape, from initial to "final" morphology, are rare – not only for individual landforms, but also for a collective "population" of spatially related landforms. As Church and Mark (1980) advised: "The most appealing avenue for resolution of the problem, in general, appears to lie in recourse to physical models. Empirical proportional relations take on a crucial role in this strategy, for they will tell us whether or not scale distortion (allometry) occurs between various combinations of the extensive properties of the prototype and model." Here, we use results from an experimental coastal barrier to demonstrate not only the emergence of collective allometric scaling relationships from spatially related washovers, but also dynamic allometry in individual washovers as they take shape.

Scaling laws are a consequence, not a direct measure, of physical process. The geomorphology literature includes plenty of remonstrations against the "blind" use of empirical scaling relationships as a kind of codex. In an essay written late in his career, geomorphologist J. Hoover Mackin remarked that "…equations read from the graphs or arrived at by other mechanical manipulations of the data are presented as terminal scientific conclusions. I suggest that the equations may be terminal engineering conclusions, but, from the point of view of science, they are statements of problems, not conclusions. A statement of a problem may be very valuable, but if it is mistaken for a conclusion, it is worse than useless because it implies

that the study is finished when in fact it is only begun" (Mackin, 1963). Making a related argument that an allometric relationship is more interesting for what it frames than what it is unto itself, Bull (1977) offered: "Allometric change is not the mere presentation of regression analyses. It is a conceptual framework for the analysis of landforms that may allow one to better understand the static and dynamic interrelations between variables that tend, or do not tend, toward equilibrium".

A strict conclusion to draw from the abstraction between the geomorphic scaling laws we can observe and the transport laws they imply is that scaling laws are "of scientific interest only if they can provide insight into the underlying mechanisms" (Church and Mark, 1980). Inasmuch as scaling laws for geomorphic features (as opposed to forces) are a manifestation of geopatterns – intrinsic spatial patterns that arise in landscapes – then such insight into underlying mechanisms speaks to some of the stated grand challenges of Earth-surface processes (NRC, 2010): How do geopatterns on Earth's surface arise, and what do they tell us about process? How do local interactions give rise to extensive, organised landscape patterns? What does spatial organisation tell us about underlying processes? And beyond those questions, what are the transport laws that govern the evolution of the Earth's surface?

Morphometric scaling laws may be more useful for approaching these grand challenges than they might seem – and nowhere more directly than in geomorphic experiments. Noting "the observed consistency between experimental and field systems despite large differences in governing dimensionless numbers," Paola et al. (2009) discuss the underappreciated power of "external similarity", in which "a small copy of a system is similar to the larger system" – even if the internal physical forces that shape the former are irreconcilably different from those that shape the latter. Paola et al. (2009) argue, as have others since (Kleinhans et al., 2014; Baynes et al., 2018), that experiments that do not conform to the rules of dynamical scaling are in fact the only way to find and test the boundaries of scale dependence and independence. A small, modelled system that looks and acts like its larger, real system might not be governed by the same transport laws, but it will convey vital information about other scaling relationships that do and do not break (van Dijk et al., 2012; Kleinhans et al., 2015; Sweeney et al., 2015; Bufe et al., 2016; Lai et al., 2016; Lazarus, 2016). This advantage is philosophically related to Bull's (1977) interest in using scaling laws to reveal "variables that tend, or do not tend, toward equilibrium".

**5 Conclusion**

In coastal settings, especially, novel methods for measuring hydrodynamics and sediment transport under storm conditions are bringing field, experimental, and numerical studies into ever better alignment (Leatherman, 1976; Leatherman and Zaremba, 1987; Matias et al., 2010; Sherwood et al., 2014; Englestad et al., 2018; Splinter et al., 2018; Phillips et al., 2019; Simmons et al., 2019; Vos et al., 2019; Wiggins et al., 2019; Dodet et al., 2019; Wesselman et al., 2019). Within the frame of geomorphology's grand challenges, such advances make dynamic coastlines "process 'hot spots' – areas where a high level of activity concentrated in a small location can be identified from relatively simple morphologic measures", especially when

"topographically based estimates…provide field scientists with a set of reference values for key local variables that serve as a starting template for observation" (NRC, 2010). Morphometric scaling relationships can be used to test numerical morphodynamic models (Lesser et al., 2004; Roelvink et al., 2009) to either confirm modelled outputs or identify areas for 365 improvement. Our work thus reiterates the utility of morphometric allometry as a window into dynamical behaviour, especially for geomorphic phenomena – such as those formed during extreme forcing events – that still confound direct observation.

**Data Availability** – The data for experimental and real washover morphology presented in this work are freely available via *figshare* (experimental and Core Banks data: doi:10.6084/m9.figshare.10259846; Ria Formosa data: doi: 10.6084/m9.figshare.10281902). The global compilation of washover data plotted in Figure 1 ("Hudock compilation") are available in an appendix of the master's thesis by Hudock (2013) and described in Hudock et al. (2014).

**Author Contributions** – EDL conceived the idea, performed the experiment, and conducted the analysis. KD measured 375 morphometry in the experiment imagery and contributed to the analysis. AM provided the Ria Formosa dataset and contributed to data interpretation. EDL drafted the manuscript, with contributions from AM and KD.

**Competing Interests** – The authors declare that they have no conflicts of interest.

**Acknowledgements** – EDL thanks Chris Paola for his invitation to conduct the original experiment at St. Anthony Falls Laboratory (SAFL) through the National Center for Earth-surface Dynamics Program (NCED2). EDL also thanks Chris 380 Ellis, Jim Tucker, Aaron Bufe, Colin Phillips, Ajay Limaye, and SAFL's student work crew for advice and assistance during the experimental trials; Evan Goldstein, Vaughan Voller, Alex Densmore, Sebastian Castelltort, Dylan McNamara, Scott Armstrong, Alida Payson, and Taylor Perron for helpful discussions; and Stuart McLelland for an invitation to the 4th Hydalab+ Workshop Event, "Advanced Workshop on Scaling Morphodynamics in Time" (January, 2018; Grenoble, France). This work was supported by the NCED2 Visiting Scientist program (via NSF grant EAR-1246761), by a research grant from 385 the British Society for Geomorphology, by the NERC BLUEcoast project (NE/N015665/2), by the EPSRC Vacation Bursary Scheme (via the University of Southampton), and by the Leverhulme Trust (RPG-2018-282). The experimental concept began on a chalkboard at the 2011 NCED Summer Institute for Earth-surface Dynamics (via NSF grant EAR-0120914). AM acknowledges the EVREST project (PTDC/MAR-EST/1031/2014), funded by the Portuguese national funding agency for science, research and technology, FCT (Fundação para a Ciência e a Tecnologia).

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

**(a)** barrier washover – field example

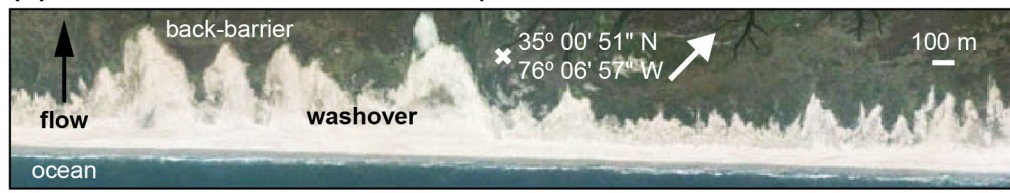

**(b)** barrier washover – experimental example

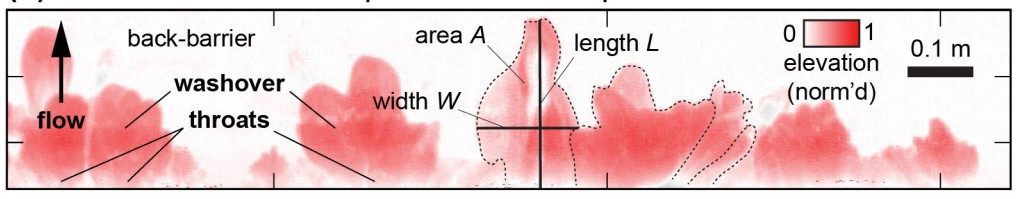

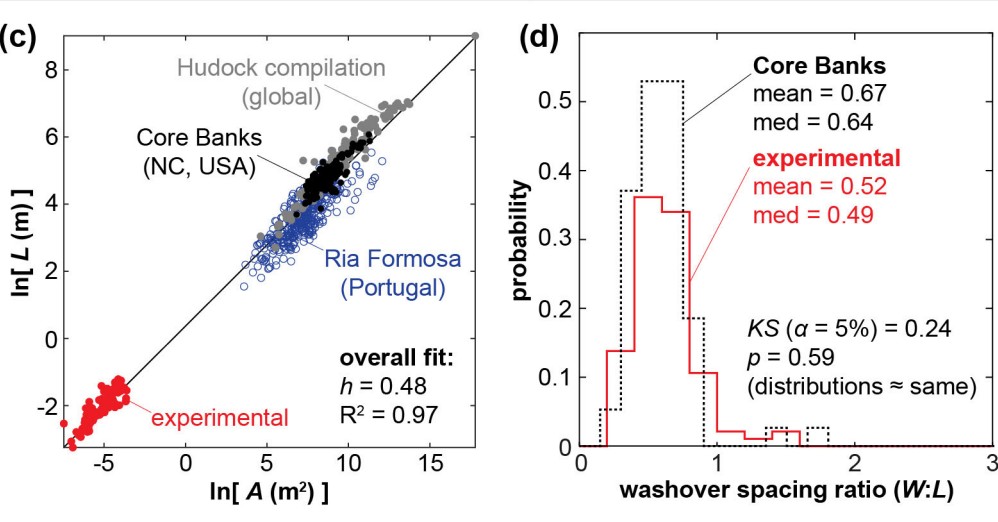

**Figure 1: This study is motivated by previous work by Lazarus (2016), from which these four panels are adapted, on morphometric relationships in experimental and real coastal overwash morphology. (a) Example of coastal barrier washover morphology on Core Banks (North Carolina, USA; image from 2015, via Google Earth). (b) Detail from a topographic laser scan (normalized relative to maximum elevation) of experimental washover morphology, annotated with key morphometric attributes (cross-shore length *L*, alongshore width *W*, area *A*). (c) Log-transform relating length and area in experimental (red) and real**
**washover deposits, including measurements from Core Banks (black dots; Lazarus, 2016), a global sample of washover depoits (grey dots; Hudock 2013; Hudock et al. 2014), and from Ria Formosa, Portugal (blue circles; Matias et al., 2008), examined in this**

study. Collectively, the datasets demonstrate a power relationship with a scaling exponent *h* = 0.48. (d) Stair plot from Lazarus (2016) comparing the alongshore spacing ratios (*W:L*) of real (Core Banks) and experimental washover.

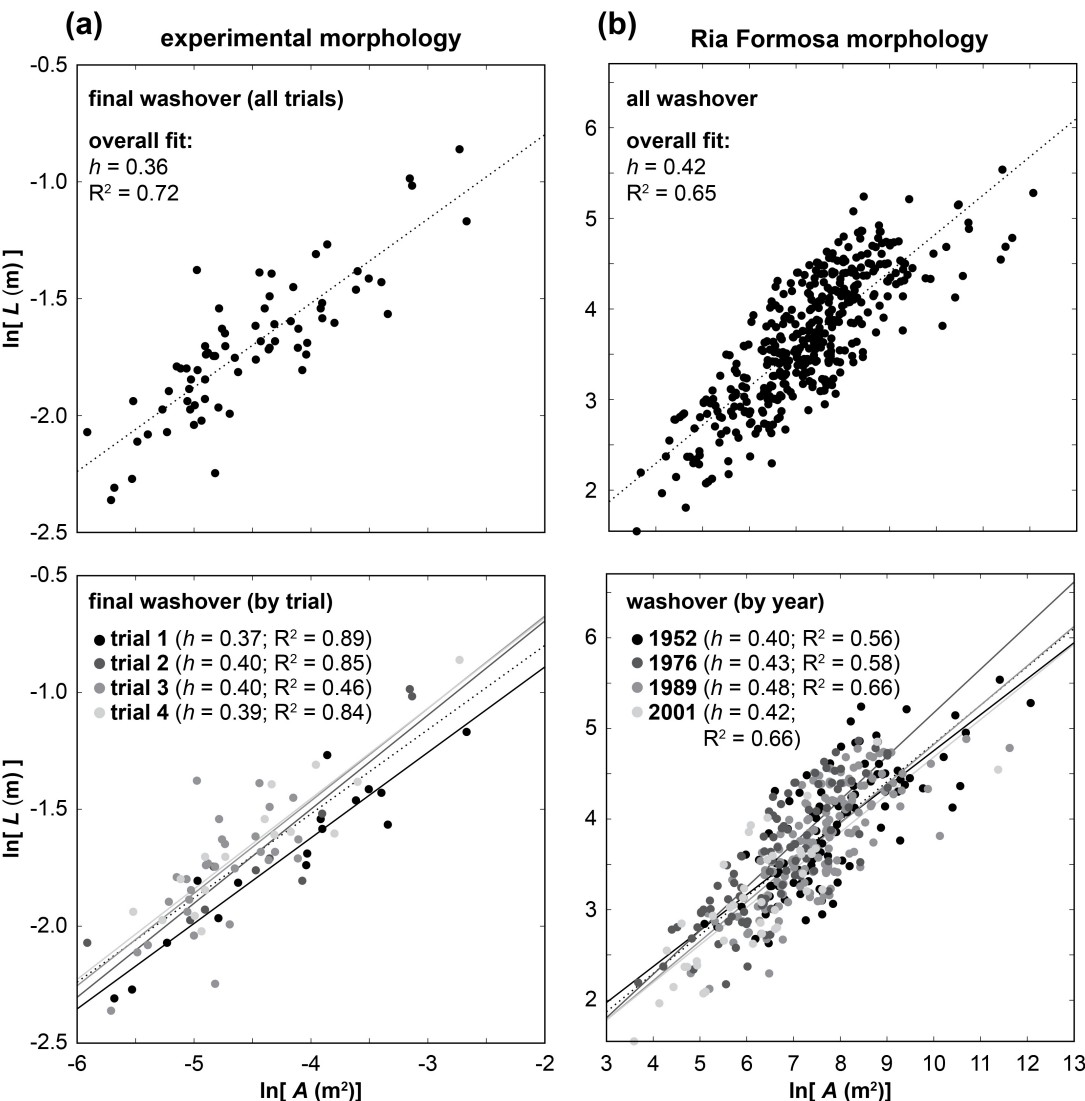

Figure 2: Log-transform comparison of length–area relationships in (a) "final", steady-state experimental washover morphology, measured from overhead imagery of the experimental tank, and (b) washovers along the Rio Formosa barriers of southern Portugal, measured from four sets of aerial images (taken in 1952, 1976, 1989, and 2001), detailed in Matias et al. (2008). Upper plots show all the data points, fitted with a power relationship. Lower plots differentiate the data according to experimental trial (left) and image year (right), each fitted with its own power relationship (along with the ensemble fits from the upper plots), demonstrating that no single trial or image year dominates the overall pattern of the data – nor does any single barrier or barrier aspect, in the Ria Formosa data (see Fig. S1). Note that the negative values for the experimental washover shown in (a) are the result of the log-transformation. Experimental and Ria Formosa data are shown on the same plot in Figure 1c.

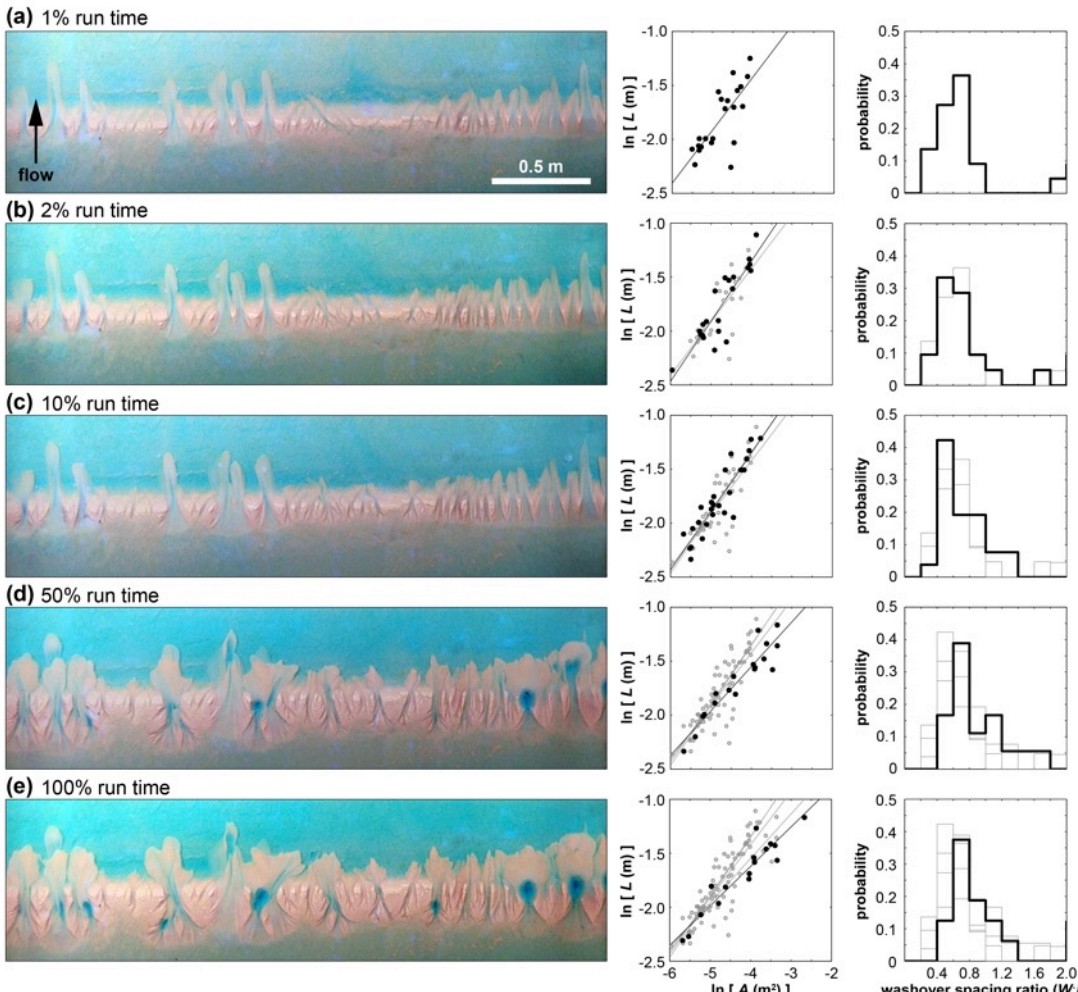

**Figure 3: A demonstration of dynamic allometry in an experimental trial of overwash morphodynamics. Time is expressed as a percentage of the total run time; 100% run time reflects the "final", steady-state morphology. Overhead images of the experimental tank (left column) show, in snapshots, the washover morphology evolving as the trial progresses (a–e). Log-transform plots of washover length relative to area (middle column) show how the scaling exponent *h* changes through time. Data points and the fitted power relationship for a given image are plotted in black; data from previous snapshots are retained in grey. Stair plots (right column) track the related development of a preferred spacing ratio (calculated as *W:L*), where the distribution at each snapshot is shown in black, and distributions from previous snapshots are retained in grey. Summary statistics are provided in Table 1.**

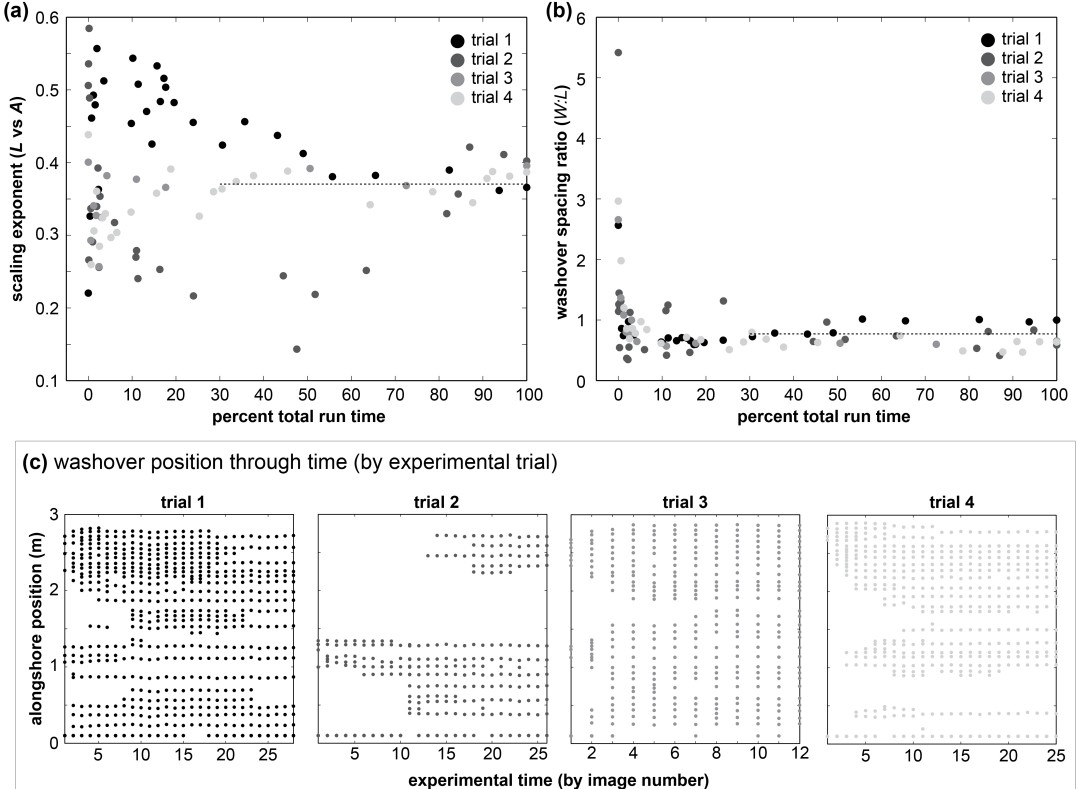

**Figure 4: Evolution of a (a) mean scaling exponent *h* and (b) preferred spacing ratio (*W:L*) compiled from four experimental trials of overwash morphodynamics (detailed in Fig. 3). Dotted lines in (a) and (b) show respective ensemble means calculated between 20% and 100% run time. Panels in (c) show progressive washover position in each of the four experimental trials. To emphasize how washover positions shift laterally before finding a morphological steady-state – the preferred spacing shown in (b) – the bottom axis reflects image sequence rather than absolute time. In (a), as in Fig. 2a, these experimental data converge on a scaling exponent smaller than the global fit (*h* = 0.48) in Fig. 1c, which spans several orders of magnitude.**

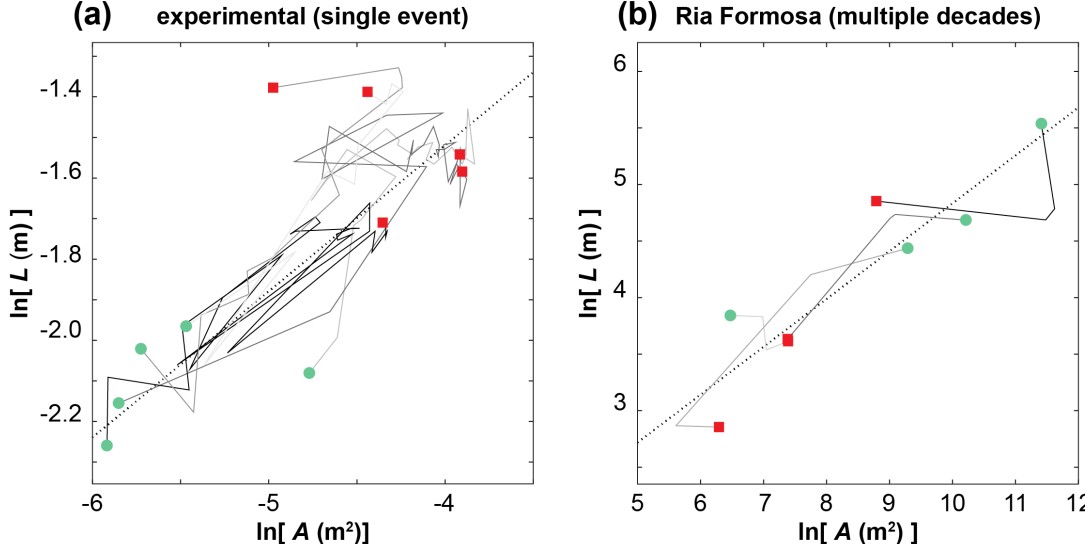

**Figure 5: Dynamic allometry in a sample of individual washovers tracked (a) during the physical experimental trials, representing a single forcing event (n = 5), and (b) identified in the four decades of aerial photographs from Ria Formosa (n = 4). Circles and squares indicate first and last measurements in the sequence, respectively. Individual trajectories are differentiated by shade (greyscale). Fits are ensemble means from Fig. 3, and listed in Table 1.**

**Table 1: Scaling exponents and distribution statistics.**

| figure & panel | data series | scaling exponent $h$ (95% confidence interval) | $R^2$ | mean spacing ratio ($W$:$L$) | median spacing ratio ($W$:$L$) |
|---|---|---|---|---|---|
| 1c | overall fit | 0.48 (0.48, 0.49) | 0.97 | | |
| 1d | Core Banks | | | 0.67 | 0.64 |
| 1d | experimental | | | 0.52 | 0.49 |
| | | | | | |
| 2a | experimental (all)* | 0.36 (0.31, 0.41) | 0.72 | | |
| 2a | trial 1 | 0.37 (0.29, 0.44) | 0.89 | | |
| 2a | trial 2 | 0.40 (0.26, 0.54) | 0.85 | | |
| 2a | trial 3 | 0.40 (0.22, 0.57) | 0.46 | | |
| 2a | trial 4 | 0.39 (0.29, 0.48) | 0.84 | | |
| 2b | Ria Formosa (all) | 0.42 (0.39, 0.45) | 0.65 | | |
| 2b | 1952 set | 0.40 (0.33, 0.48) | 0.56 | | |
| 2b | 1976 set | 0.43 (0.38, 0.49) | 0.58 | | |
| 2b | 1989 set | 0.48 (0.41, 0.55) | 0.66 | | |
| 2b | 2001 set | 0.42 (0.32, 0.51) | 0.66 | | |
| | | | | | |
| 3a | trial 1 @ 1% run time | 0.49 | | 0.74 | 0.51 |
| 3b | trial 1 @ 2% run time | 0.56 | | 0.80 | 0.52 |
| 3c | trial 1 @ 10% run time | 0.54 | | 0.62 | 0.57 |
| 3d | trial 1 @ 50% run time | 0.41 | | 0.79 | 0.65 |
| 3e | trial 1 @ 100% run time | 0.37 | | 1.0 | 0.69 |
| | | | | | |
| 4a | mean (30–100% time) | 0.37 | | | |
| 4b | mean (30–100% time) | | | 0.77 | |
| | | | | | |
| 5a | experimental (from 2a) | 0.36 | | | |
| 5b | Ria Formosa (from 2b) | 0.42 | | | |
| | | | | | |

**\* Note: Lazarus (2016) reported an ensemble fit (for final morphology, based on topographic laser scans) of $h$ = 0.48 (0.43, 0.53; $R^2$ = 0.77).**