# Peer review of "Dynamic allometry in coastal overwash morphology"

_Earth Surface Dynamics, 2019_

## Referee Comment (RC1) · Katherine Ratliff (Referee) · 28 Aug 2019

Lazarus et al describes how dynamic allometry manifests in washover deposits from a set of laboratory experiments simulating barrier island inundation and shows how these scaling relationships compare to five decades of washover deposit imagery from the Ria Formosa barrier system. The content of the manuscript is interesting and wellwritten. I found the similarities in scaling relationships between the experimental and natural washover deposits remarkable, especially given the simplicity of the experimental setup relative to the natural setting (e.g., lack of dunes, vegetation, back-barrier marsh, etc). Although these findings are discussed, I did feel like the somewhat lengthy and more general discussion of scaling laws in the Implications section, albeit interesting, diluted what could be a more impactful presentation of the study's experimental

and comparative results. Perhaps some of this information is better suited in the Introduction, or including a conclusions section that more explicitly addresses the work presented here, would better highlight the manuscript's scientific contribution. Specific comments are listed below.

II. 17-19: Last sentence of abstract – the importance of initial conditions does not appear to be a focus of the manuscript

I. 79: Does this mean that the barrier width varied alongshore or between trials?

Section 2.2: Some more general information about the Ria Formosa barrier system (e.g., average dimensions, how densely vegetated, average overwash/inundation frequency) would be useful for comparison with the experimental washover setup.

II. 142-144: These two sentences are generally true, but there are also marked gaps between washover deposits that persist over the course of the experiments, particularly in trial 2 (Fig. 4c). Can you address why this irregular spacing can also be seemingly stable over time? And could this partly be the cause of differences between the trials' dynamic allometry, seen in Fig. 4a?

II. 156-158: Couldn't allogenic factors also affect washover deposit morphology over the time scales of multiple decades, e.g., relative sea-level rise, erosion of the back-barrier marsh, changing sediment supply, etc.?

II. 226-227: Sentence needs rewording

II. 234-236: This excerpt from Perron and Fagherazzi (2012) is referencing different landscape features tending towards equilibrium states, i.e., comparing a drainage divide vs. valley arrangement. Here, only one landscape feature (washover) is being considered, and although these features are in different stages towards equilibrium, the comparison is not entirely clear since it is one type of feature.

I. 281: suggest replacing "to" with "we"

**ESurfD**
**ESurfD**

---

## Referee Comment (RC2) · Anonymous Referee #2 · 4 Sep 2019

The authors utilize data from a set of laboratory experiments and a field case over several decades to document the dynamic allometry of overwash deposits. In particular, they show that from the collective growth (length and area) of the overwash deposits the final allometric relations emerge. In general, I enjoyed reading this manuscript and think it would fit within the scope of Earth Surface Dynamics. I appreciate the shorter length as well. The paper is well written and easy to read, however it is not always easy to follow the narrative. As far as I can tell I see no technical issues with the methods and results, however I recommend some revision related to narrative issues prior to publication to address a couple aspects. (1) It is not immediately clear what the utility of the results are and how one might use these going forward, in that being a non-expert on coastal morphodynamics the next steps are not apparent to me. (2) It

feels like there is a narrative disconnect between the more philosophical elements and the analysis presented here. This discussion is interesting, but it also detracts from the results as these two elements don't quite sync up (more on this below).

A potential drawback of the manuscript in its current form is that it feels like there are two narratives throughout the paper. One on the allometry of overwash deposits, and the other is more philosophical. They don't quite come together, at the end I am left wondering what I learned about overwash deposits and how I might use this information going forward. As a reader I would appreciate more discussion on the particular datasets analyzed here and what this tells us about coastal barriers. An example of the two narratives is the transition between the results and the discussion section is a bit abrupt with the discussion of erosional mountain valleys and feels a little far afield from the methods and results section.

I would appreciate it if the authors could add a paragraph on how future research might use their results. Along these lines, it would be useful to add some discussion on what novel processes might have emerged from tracking the dynamic allometry of overwash deposits.

Specific Comments. Abstract. This is a nicely written abstract, but it is not completely clear what the results of the paper are. Some additional details on the analysis and conclusions would be welcome within the abstract. For example, could you be a little more specific about what the different patterns of change over longer time scales are? It would also help to add a line showing what the initial conditions were and what they became, this would help with the final sentence which is a bit broad at the moment.

Figure 1. Could you add the Ria Formosa data to 1a? This would be helpful as the experimental and Formosa data are never compared on the same plot.

Ln. 75. Should the parenthetical statement be inside the previous period?

Ln. 103. Consider including a photograph of the field site.

Ln. 136. Could you comment on what causes the spacing converges at a faster rate than the slope to the length and area relation?

Figure 4a. Could you add an explanation as to why the final exponent is smaller than the observed global fit in Figure 1? Figure 4c. I am not quite sure what I should be taking away from this figure. Maybe add a line to the caption on what the reader should be focusing on in here.

Ln. 165-175. This (along with the patterns in Fig. 5 a & b) is really interesting. Is there evidence for the Ria Formosa to constrain why they (on average) got smaller over time? The discussion on vegetation growth is interesting, but it is not clear if the authors think this happened within their data.

Acknowledgments. I didn't see a data availability statement.

---

## Referee Comment (RC3) · Anonymous Referee #3 · 16 Sep 2019

The authors assess dynamic allometry relations of barrier systems overwash morphology by analyzing a set of vertical images from lab experiments and comparing the results with field imageries of Ria Formosa barriers in Portugal. I think the work is important and the results are interesting. I just have some minor suggestions that the paper could benefit from.

I strongly recommend separating the methods from sections 2 and 3 and having it under a new section as "Methods". Also, there is some useful discussion about vegetation in 3.2 that is more related to the Discussion section. Reconstructing the text for this would ease the reading.

There is no justification why the parameters used in the experiment (e.g., barrier height and infill rate) are reasonable and the results can be comparable with real case scenarios. Also, there is no discussion on limitations of the experiments and uncertainty of the experimental results.

Line 25: adding an example of allometry would be helpful.

Line 52: switch the order of examples to match the order of static allometry types in the previous lines.

Line 78: How can the size of the sediment used in the experiment affect the results?

Line 181: Briefly describe what Mosely and Parker (1972) work is about, what they do for those who are not familiar with their work.

Line 227: remove extra "the"

The last sentence of the first paragraph of Results is out of context.

Add h and R2 values in fig 2.

Lastly, I did not find it very useful to quote from many other literature in the second half of the paper. It was very confusing and I had to read the sentences few times to understand the points.

---

## Author Comment (AC1) · 1 Oct 2019

Please find our response in the linked PDF.

Please also note the supplement to this comment:
https://www.earth-surf-dynam-discuss.net/esurf-2019-39/esurf-2019-39-AC1-supplement.pdf
* * *

---

## Author Comment (AC2) · 1 Oct 2019

**ESURFD-2019-39 (Lazarus, Davenport & Matias)**

**Preliminary Reply to R2**

Reviewer comments in *italics*; authors' preliminary reply in **bold**.

*The paper is well written and easy to read, however it is not always easy to follow the narrative. As far as I can tell I see no technical issues with the methods and results, however I recommend some revision related to narrative issues prior to publication to address a couple aspects. (1) It is not immediately clear what the utility of the results are and how one might use these going forward, in that being a non-expert on coastal morphodynamics the next steps are not apparent to me.*

**Noted. We can add an explanation that these sorts of results can be fed into numerical models as probability-distribution functions that may inform and/or guide model behaviour. That may be the most immediate utility. From those models, predictions (or at least forecasts) can be made about potential washover magnitudes. That information is relevant to the emergency-response crews (among others) who clear roadways, and, more generically, to the estimation of barrier sediment budgets used by engineers, geologists, and ecologists.**

*(2) It feels like there is a narrative disconnect between the more philosophical elements and the analysis presented here. This discussion is interesting, but it also detracts from the results as these two elements don't quite sync up (more on this below).*

*A potential drawback of the manuscript in its current form is that it feels like there are two narratives throughout the paper. One on the allometry of overwash deposits, and the other is more philosophical. They don't quite come together, at the end I am left wondering what I learned about overwash deposits and how I might use this information going forward. As a reader I would appreciate more discussion on the particular datasets analyzed here and what this tells us about coastal barriers. An example of the two narratives is the transition between the results and the discussion section is a bit abrupt with the discussion of erosional mountain valleys and feels a little far afield from the methods and results section.*

**Understood. This comment aligns with a similar remark by R#1, who has recommended changes to the Introduction, Implications, and possible Conclusions sections to achieve, a more impactful presentation of the study' findings. We will make sure these changes likewise reconcile the apparently parallel narratives of the manuscript. R#1 has also recommended that we clarify the section on ridge-and-valley topography – which we can make clear is a surprisingly cognate system.**

*I would appreciate it if the authors could add a paragraph on how future research might use their results. Along these lines, it would be useful to add some discussion on what novel processes might have emerged from tracking the dynamic allometry of overwash deposits.*

**As above – we can clarify the potential utility of data like these.**

*Specific Comments. Abstract. This is a nicely written abstract, but it is not completely clear what the results of the paper are. Some additional details on the analysis and conclusions would be welcome within the abstract. For example, could you be a little more specific about what the different patterns of change over longer time scales are? It would also help to add a line showing what the initial conditions were and what they became, this would help with the final sentence which is a bit broad at the moment.*

**Noted – will revise. R#1 also commented on the mention of initial conditions in the Abstract.**

*Figure 1. Could you add the Ria Formosa data to 1a? This would be helpful as the experimental and Formosa data are never compared on the same plot.*

**Noted – will consider and revise.**

*Ln. 75. Should the parenthetical statement be inside the previous period?*

**Noted – will revisit.**

*Ln. 103. Consider including a photograph of the field site.*

**Noted – will consider.**

*Ln. 136. Could you comment on what causes the spacing converges at a faster rate than the slope to the length and area relation?*

**Interesting observation – will explore and amend.**

*Figure 4a. Could you add an explanation as to why the final exponent is smaller than the observed global fit in Figure 1? Figure 4c. I am not quite sure what I should be taking away from this figure. Maybe add a line to the caption on what the reader should be focusing on in here.*

**Noted – will revise for clarity.**

Ln. 165-175. This (along with the patterns in Fig. 5 a & b) is really interesting. Is there evidence for the Ria Formosa to constrain why they (on average) got smaller over time? The discussion on vegetation growth is interesting, but it is not clear if the authors think this happened within their data.

**Noted – will explore and amend.**

*Acknowledgments. I didn't see a data availability statement.*

**A data-availability statement will come with the published paper.**

---

## Author Comment (AC3)

**ESURFD-2019-39 (Lazarus, Davenport & Matias)**

**Preliminary Reply to R3**

Reviewer comments in *italics*; authors' preliminary reply in **bold**.

*I just have some minor suggestions that the paper could benefit from.*

*I strongly recommend separating the methods from sections 2 and 3 and having it under a new section as "Methods". Also, there is some useful discussion about vegetation in 3.2 that is more related to the Discussion section. Reconstructing the text for this would ease the reading.*

**We will consider and revise for clarity. Reviewers 1 & 2 have both recommended amendments to the Introduction and Discussion that will have bearing on this suggestion.**

*There is no justification why the parameters used in the experiment (e.g., barrier height and infill rate) are reasonable and the results can be comparable with real case scenarios. Also, there is no discussion on limitations of the experiments and uncertainty of the experimental results.*

**The experiment is described in full in Lazarus (GRL, 2016) and is a generic "analogue" model with no explicit connection to real scenarios in the sense of direct simulation. The scaling behaviour of the morphology, not the parameters, makes the results comparable to real cases. Regardless, we can use this comment to find ways to clarify the experimental description.**

*Line 25: adding an example of allometry would be helpful.*

**Noted – will amend.**

*Line 52: switch the order of examples to match the order of static allometry types in the previous lines.*

**Noted – will amend.**

*Line 78: How can the size of the sediment used in the experiment affect the results?*

**Larger sediment will likely make blunter deposits; finer sediment will tend to make more finger-like deposits. We did not test this directly, but there are reasonable examples from the field to mention – we will consider and revise.**

*Line 181: Briefly describe what Mosely and Parker (1972) work is about, what they do for those who are not familiar with their work.*

**Noted – will clarify and revise.**

*Line 227: remove extra "the"*

**Noted – will correct.**

*The last sentence of the first paragraph of Results is out of context. Add h and R2 values in fig 2.*

**Noted – will amend.**

*Lastly, I did not find it very useful to quote from many other literature in the second half of the paper. It was very confusing and I had to read the sentences few times to understand the points.*

**Noted – we will revisit this stylistic choice.**

---

## Author Response (AR1)

**RESPONSE TO REVIEWS**

**ESURFD-2019-39 (Lazarus, Davenport & Matias)**

Dear Editors –

Many thanks for your continued interest in this manuscript, and to the three reviewers for their time and attention. Please find attached a revised draft of the work. This document provides a point-by-point response to the reviewers' comments.

Reviewer comments in *italics*; authors' reply in **bold**; new text in blue.

Sincerely –

Eli Lazarus (*et alia*; *E.D.Lazarus@soton.ac.uk*)
* * *
**Reply to R1 (Ratliff):**

*I found the similarities in scaling relationships between the experimental and natural washover deposits remarkable, especially given the simplicity of the experimental setup relative to the natural setting (e.g., lack of dunes, vegetation, back-barrier marsh, etc). Although these findings are discussed, I did feel like the somewhat lengthy and more general discussion of scaling laws in the Implications section, albeit interesting, diluted what could be a more impactful presentation of the study's experimental and comparative results. Perhaps some of this information is better suited in the Introduction, or including a conclusions section that more explicitly addresses the work presented here, would better highlight the manuscript's scientific contribution.*

**We have made significant adjustments across the bookending sections of the manuscript to better frame and articulate the study's findings – a recommendation also raised by R2.**

*ll. 17-19: Last sentence of abstract – the importance of initial conditions does not appear to be a focus of the manuscript*

**We have revised the Abstract (L16–20) to read:**

"…we find differences between patterns of morphometric change at the event scale versus longer time scales. Our results may help inform and test process-based morphodynamic coastal models, which typically use statistical distributions and scaling laws to underpin empirical or semi-empirical parameters at fundamental levels of model architecture. More broadly, this work dovetails with theory for landscape evolution more commonly associated with fluvial and alluvial terrain, offering new evidence from a coastal setting that a landscape may reflect characteristics associated with an equilibrium or steady-state condition even when features within that landscape do not."

*l. 79: Does this mean that the barrier width varied alongshore or between trials?*

**Revised – the barrier width did not vary alongshore; for the trials presented, the stated range is unnecessary.**

*Section 2.2: Some more general information about the Ria Formosa barrier system (e.g., average dimensions, how densely vegetated, average overwash/inundation frequency) would be useful for comparison with the experimental washover setup.*

**Added text at L142–146 to read:**

Washover length (cross-shore distance between barrier crest and back-barrier edge) in the Ria Formosa data reached a maximum of 250 m. Barrier morphology varies substantively within the Ria Formosa system, ranging by an order of magnitude in island width (in some cases along the same island), with differing patterns and extents of dune fields, urban footprint, and proximities to mesotidal inlets. Here we examine the Ria Formosa not because of any direct correspondence to the barrier design in the physical experiments, but because the system offers a closely examined source of repeated measurements of persistent washover footprints along its ~60 km spatial extent.

*ll. 142-144: These two sentences are generally true, but there are also marked gaps between washover deposits that persist over the course of the experiments, particularly in trial 2 (Fig. 4c). Can you address why this irregular spacing can also be seemingly stable over time? And could this partly be the cause of differences between the trials' dynamic allometry, seen in Fig. 4a?*

**Added text at L193–211 to read:**

Also evident in the experimental results are irregular, persistent gaps between washover sites (Fig. 4c), which are likely a consequence of overwash flow competition and partitioning. The infill rate was never varied, meaning the hydrodynamic forcing was held constant throughout each trial. Moreover, the surface of the upstream, "ocean-side" reservoir was never perturbed (agitated with a wave paddle, for example). This means that once enough overwash breaches had formed to accommodate and distribute the forcing flow across the barrier, new breaches were either unlikely to develop, or would only develop as a consequence of subtle, local interactions between adjacent overwash throats. For example, if flow through a throat slowed down, sediment caught in the throat could form a temporary plug. That plug appeared to drive a backwater effect that elevates the upstream water surface just enough to force the overwash flow toward a new path of steepest descent – typically down through a neighboring throat, but also over an otherwise undissected reach of the barrier. (These plug-and-backwater dynamics were observed, but not measured directly.) Over many tens of minutes, the overwash morphology adjusted to flow conditions and the elevation difference between the barrier top and back-barrier plane, sets of neighbouring throats might plug and unplug several times, with corresponding periods of dormancy or growth in their associated washover deposits. Given that they were subject to the same forcing conditions, all breaches in the barrier should have tended to adjust toward the same open-channel geometry. However, because throats had to share – compete for – available overwash flow, closely spaced sets of throats grew more slowly than an isolated throat with no nearby neighbors. In

a natural case, flow-limited conditions may mean that for a series of overwash throats, no single throat may ever capture enough flow to reach its equilibrium open-channel configuration. Storm-driven water levels in the field (Shaw et al., 2015; Englestad et al., 2018; Wesselman et al., 2019) may rise and fall much faster than the time scales required for overwash morphology to reach a geometric equilibrium.

*ll. 156-158: Couldn't allogenic factors also affect washover deposit morphology over the time scales of multiple decades, e.g., relative sea-level rise, erosion of the back- barrier marsh, changing sediment supply, etc.?*

**Revised text at L240–249 to read:**

Furthermore, gradual processes of vegetation recovery and aeolian sand deposition within the washover can progress in irregular ways related to natural topographic heterogeneity in barrier and dune morphology, forcing the washover to inherit morphometric characteristics dynamically unrelated to overwash and inundation processes (Morton and Sallenger, 2003; Matias et al., 2008). Scaling controls likely manifest in conjunction with, not in place of, other allogenic factors that affect overwash morphology over multi-decadal time scales, such as relative sea-level rise, changes in shoreline position and sediment supply, and heterogeneity in shoreface lithology. Preferred geometric relationships in overwash morphology may both set a template for smaller-scale, faster-forming barrier features and be forced to conform to contextual controls exerted by barrier-scale geography (Werner, 2003; Coco and Murray, 2007).

*ll. 226-227: Sentence needs rewording*

**Moved sentence and revised to read (L300–302):**

The dynamic allometry of individual washovers that we record (Fig. 5a) also bears on a broader geomorphological question regarding landscape convergence toward an equilibrium state – and, in particular, how such convergence is physically expressed (Bull, 1975; Perron and Fagherazzi, 2012).

*ll. 234-236: This excerpt from Perron and Fagherazzi (2012) is referencing different landscape features tending towards equilibrium states, i.e., comparing a drainage divide vs. valley arrangement. Here, only one landscape feature (washover) is being considered, and although these features are in different stages towards equilibrium, the comparison is not entirely clear since it is one type of feature.*

**With the addition of the new paragraph at L193–211 and an emphasis (L260) that the regular spacing extends in the alongshore dimension, this comparison to drainage divides should now be clearer.**

*l. 281: suggest replacing "to" with "we"*

**Amended sentence.**

*Fig 4: reference to Fig. 2b in caption seems incorrect*

**Amended caption.**

**Reply to R2:**

*The paper is well written and easy to read, however it is not always easy to follow the narrative. As far as I can tell I see no technical issues with the methods and results, however I recommend some revision related to narrative issues prior to publication to address a couple aspects. (1) It is not immediately clear what the utility of the results are and how one might use these going forward, in that being a non-expert on coastal morphodynamics the next steps are not apparent to me.*

**We have added several contextual elements to the Introduction to explain how and why this line of inquiry is relevant. For example, at L17– 19 of the Abstract (and again later in the manuscript), we point out that:**

Our results may help inform and test process-based morphodynamic coastal models, which typically rely on empirical or semi-empirical parameters like scaling laws at fundamental levels of their architecture.

**At L60–62, we explain why understanding allometry matters:**

Linking the static allometry of a given shape to its dynamic allometry – observing the progression by which a form emerges – is fundamental to linking overarching pattern to underlying process.

**And at L76–84, we explain the importance of understanding overwash as a physical coastal process:**

Overwash is a natural, fundamental physical process of coastal barrier systems in which shallow overland flow, driven by a storm event, transports sediment from the open-coastal barrier face to the barrier floodplain and sheltered back-barrier wetlands (Morton and Sallenger, 2003; Donnelly et al., 2006; FitzGerald et al., 2008). Overwash occurs even in the absence of sea-level rise, but sediment supply to the floodplain and back-barrier environments through washover deposition is the vital mechanism that allows barrier systems to maintain elevation and width relative to sea level over time scales of centuries to millennia (FitzGerald et al., 2008). Although essential to coastal barrier evolution (and, by extension, to the function of natural barrier ecosystems), overwash becomes a hazard where it interacts with coastal infrastructure and built environments (Rogers et al., 2015; Lazarus and Goldstein, 2019). Overwash morphology is thus at the crux of understanding current – and anticipating future – coastal environments and risk along low-lying open coastlines (Wong et al., 2014).

*(2) It feels like there is a narrative disconnect between the more philosophical elements and the analysis presented here. This discussion is interesting, but it*

*also detracts from the results as these two elements don't quite sync up (more on this below).*

*A potential drawback of the manuscript in its current form is that it feels like there are two narratives throughout the paper. One on the allometry of overwash deposits, and the other is more philosophical. They don't quite come together, at the end I am left wondering what I learned about overwash deposits and how I might use this information going forward. As a reader I would appreciate more discussion on the particular datasets analyzed here and what this tells us about coastal barriers. An example of the two narratives is the transition between the results and the discussion section is a bit abrupt with the discussion of erosional mountain valleys and feels a little far afield from the methods and results section.*

**In light of this comment, and similar remarks from R1, we have made significant amendments to the Introduction and elements of the Results to better frame the Discussion & Implications at the end of the manuscript.**

*I would appreciate it if the authors could add a paragraph on how future research might use their results. Along these lines, it would be useful to add some discussion on what novel processes might have emerged from tracking the dynamic allometry of overwash deposits.*

**See added text regarding relevance to numerical models (L17–19; excerpted above), and L93–96:**

Given that leading process-based models of coastal morphodynamics have embedded in their architectures a host of semi-empirical parameters (Simmons et al., 2019), scaling relationships derived from static and dynamic allometry for overwash morphology may first serve tests of model predictions, en route to integration into predictive models themselves.

*Specific Comments. Abstract. This is a nicely written abstract, but it is not completely clear what the results of the paper are. Some additional details on the analysis and conclusions would be welcome within the abstract. For example, could you be a little more specific about what the different patterns of change over longer time scales are? It would also help to add a line showing what the initial conditions were and what they became, this would help with the final sentence which is a bit broad at the moment.*

**Revised Abstract (see response to R1, above).**

*Figure 1. Could you add the Ria Formosa data to 1a? This would be helpful as the experimental and Formosa data are never compared on the same plot.*

**Ria Formosa data now appears in Fig. 1c.**

*Ln. 75. Should the parenthetical statement be inside the previous period?*

**Sentence amended.**

*Ln. 103. Consider including a photograph of the field site.*

**We do not include an image of Ria Formosa in this manuscript, but Matias et al. (2008) present an exhaustive description of the area.**

*Ln. 136. Could you comment on what causes the spacing converges at a faster rate than the slope to the length and area relation?*

**In response to this interesting observation, we have added the following text (L185–191):**

If washovers are initially too far apart, they widen, and some new deposits fill in between them, until the alongshore pattern reaches a closer spacing configuration. Conversely, if washovers are initially too close, they merge, effectively adjusting their centroids to be farther apart. The spacing ratio converges on a quasi-equilibrium configuration (Fig. 4b) more quickly than the length-to-area relationship, perhaps because the mean spacing ratio is a more stable metric than the scaling exponent, which is comparatively more sensitive to the influence of larger washover deposits as they grow. That is, a large washover deposit is less likely to markedly shift the centre of a univariate distribution of spacing ratios than it is to affect a best-fit power relationship between length and area.

*Figure 4a. Could you add an explanation as to why the final exponent is smaller than the observed global fit in Figure 1? Figure 4c. I am not quite sure what I should be taking away from this figure. Maybe add a line to the caption on what the reader should be focusing on in here.*

**Added to Fig. 4 caption (L599–600):**

In (a), as in Fig. 2a, these experimental data converge on a scaling exponent smaller than the global fit ($h$ = 0.48) in Fig. 1c, which spans several orders of magnitude.

Ln. 165-175. This (along with the patterns in Fig. 5 a & b) is really interesting. Is there evidence for the Ria Formosa to constrain why they (on average) got smaller over time? The discussion on vegetation growth is interesting, but it is not clear if the authors think this happened within their data.

**We have clarified that this sort of vegetation effect is likely what we see in the Ria Formosa data (L231–233):**

Such dimensional adjustment could stem from depth-dependent zonation in barrier vegetation. (Matias et al. (2008) discuss the potential influence of barrier vegetation on Ria Formosa overwash morphology, but do not measure it directly.) Storm deposits can drive spatial heterogeneity in vegetation growth rates…

*Acknowledgments. I didn't see a data availability statement.*

**Now included.**

**Reply to R3:**

*I just have some minor suggestions that the paper could benefit from.*

*I strongly recommend separating the methods from sections 2 and 3 and having it under a new section as "Methods". Also, there is some useful discussion about vegetation in 3.2 that is more related to the Discussion section. Reconstructing the text for this would ease the reading.*

**We have made significant revisions to the motivation for Sections 2 & 3, and to the sequence in which both sections are laid out. We hope that these amendments have made the through-line easier to follow.**

*There is no justification why the parameters used in the experiment (e.g., barrier height and infill rate) are reasonable and the results can be comparable with real case scenarios. Also, there is no discussion on limitations of the experiments and uncertainty of the experimental results.*

**We have added the following text to L120–128:**

Washover size could be increased by increasing the barrier height. A height of 2 cm was chosen because that elevation generated approximately five times more overwash features alongshore than a barrier with height 4 cm (see Supporting Information for Lazarus, 2016). The infill rate was the maximum possible for the experimental basin, and not powerful enough to simply obliterate the initial barrier (a catastrophic storm was not the intention of this experimental design). Although grain size was not directly tested as a control on experimental overwash morphology (working to time and labour constraints, we used the sand that was already installed in the basin at the time of its availability), we inferred that a larger grain size would likely result in blunter lobes, and a finer grain size in more "finger-like" deposits (Homsey, 1987) significantly greater in cross-shore length relative to their alongshore width.

**As noted above (in response to R1), we have also added L144–146:**

Here we examine the Ria Formosa not because of any direct correspondence to the barrier design in the physical experiments, but because the system offers a closely examined source of repeated measurements of persistent washover footprints along its ~60 km spatial extent.

*Line 25: adding an example of allometry would be helpful.*

**We now offer the following, at L29–33:**

Allometric patterns in geomorphology appear in a diversity of settings – erosional (river systems, submarine canyons) and depositional (alluvial fans, coastal deltas) – and are a quantitative signature of intrinsic structure, organisation, or regularity (Church and Mark, 1980; Dodds and Rothman,

2000; Moscardelli and Wood, 2006; Straub et al., 2007; Paola et al., 2009; Wolinsky et al., 2010; Edmonds et al., 2011; Lazarus, 2016).

*Line 52: switch the order of examples to match the order of static allometry types in the previous lines.*

**Revised as suggested.**

*Line 78: How can the size of the sediment used in the experiment affect the results?*

**Larger sediment will likely make blunter deposits; finer sediment will tend to make more finger-like deposits: see new text (excerpted in response to related R3 comment, above), at L120–128.**

*Line 181: Briefly describe what Mosely and Parker (1972) work is about, what they do for those who are not familiar with their work.*

**At L48–50, we write:**

A test of drainage-network evolution in an experimental basin by Mosley and Parker (1972) did not confirm Woldenberg's (1966) argument for allometric growth: they found no evidence that rates of size-correlated changes in drainage networks conformed to a well-organised pattern.

**And at L252–254, we clarify:**

Echoing Mosley and Parker (1972), who found no clear indication of allometric growth in the evolution of an experimental river network, we find no clear indication of allometric growth in washover (Fig. S2).

*Line 227: remove extra "the"*

**Revised as suggested.**

*The last sentence of the first paragraph of Results is out of context. Add h and R2 values in fig 2.*

**Revised as suggested – and we have added $R^2$ & 95% confidence intervals to Table 1.**

*Lastly, I did not find it very useful to quote from many other literature in the second half of the paper. It was very confusing and I had to read the sentences few times to understand the points.*

**Revised as suggested.**

[revised manuscript text omitted]